# Hard Voting Ensemble Approach for the Detection of Type 2 Diabetes in Mexican Population with Non-Glucose Related Features

**DOI:** 10.3390/healthcare10081362

**Published:** 2022-07-22

**Authors:** Jorge A. Morgan-Benita, Carlos E. Galván-Tejada, Miguel Cruz, Jorge I. Galván-Tejada, Hamurabi Gamboa-Rosales, Jose G. Arceo-Olague, Huizilopoztli Luna-García, José M. Celaya-Padilla

**Affiliations:** 1Unidad Académica de Ingeniería Eléctrica, Universidad Autónoma de Zacatecas, Jardín Juárez 147, Centro, Zacatecas 98000, Mexico; alejandro.morgan@uaz.edu.mx (J.A.M.-B.); ericgalvan@uaz.edu.mx (C.E.G.-T.); gatejo@uaz.edu.mx (J.I.G.-T.); hamurabigr@uaz.edu.mx (H.G.-R.); arceojg@uaz.edu.mx (J.G.A.-O.); 2Unidad de Investigación Médica en Bioquímica, Hospital de Especialidades, Centro Médico Nacional Siglo XXI, Instituto Mexicano del Seguro Social, Av. Cuauhtémoc 330, Col. Doctores, Del. Cuauhtémoc, Mexico City 06720, Mexico; miguel.cruzlo@imss.gob.mx

**Keywords:** ensemble model, machine learning, logistic regression, support vector machine, neural networks, type 2 diabetes mellitus detection

## Abstract

Type 2 diabetes mellitus (T2DM) represents one of the biggest health problems in Mexico, and it is extremely important to early detect this disease and its complications. For a noninvasive detection of T2DM, a machine learning (ML) approach that uses ensemble classification models with dichotomous output that is also fast and effective for early detection and prediction of T2D can be used. In this article, an ensemble technique by hard voting is designed and implemented using generalized linear regression (GLM), support vector machines (SVM) and artificial neural networks (ANN) for the classification of T2DM patients. In the materials and methods as a first step, the data is balanced, standardized, imputed and integrated into the three models to classify the patients in a dichotomous result. For the selection of features, an implementation of LASSO is developed, with a 10-fold cross-validation and for the final validation, the Area Under the Curve (AUC) is used. The results in LASSO showed 12 features, which are used in the implemented models to obtain the best possible scenario in the developed ensemble model. The algorithm with the best performance of the three is SVM, this model obtained an AUC of 92% ± 3%. The ensemble model built with GLM, SVM and ANN obtained an AUC of 90% ± 3%.

## 1. Introduction

According to the World Health Organization (WHO), diabetes is “a chronic metabolic disease characterized by high levels of glucose in the blood, which leads over time to serious damage to the heart, blood vessels, eyes, kidneys and nerves” [1], being Type 2 diabetes mellitus (T2DM) the most common type. T2DM develops primarily due to an inactive lifestyle, lack of exercise, and body weight [2]. According to the International Diabetes Federation (IDF), in 2021 there were 537 million cases in the world with ages between 20 and 79 years, 541 million adults are at increased risk of developing T2DM [3]. For more than 60 million people living with T2DM, insulin is essential to reduce the risk of kidney failure, blindness and limb amputation [4]. The proportion of people affected with T2DM that is related to medical complications is alarming, either because of the lack of these drugs or because of their cost. In Mexico, according to the 2020 mortality data given by the Instituto Nacional de Estadística y Geografía INEGI, in Mexico, 1,086,743 deaths were reported, of which 14% (151,019) correspond to deaths from diabetes mellitus. Of these, 52% (78,922) occurred in men and 48% (72,094) in women. Of the total deaths, 98% (144,513) were due to non-insulin-dependent diabetes and 2% (3506) to insulin-dependent diabetes [5].

A vital approach to preventing or treating T2DM is through early diagnosis [6]. One of the most common ways to get this diagnosis is by having a blood test to check your glucose levels and then taking action based on the results. These types of tests are generally invasive, and only carried out after eminent symptoms, take some time and are very impractical and these types of tests tend to be recurrent, which implies a significant amount of money in the long term. The use of surveys to obtain clinical data from the population of a certain region has been shown to be inefficient, expensive, and most of the time, they cannot be trusted by researchers [7]. As this becomes a problem, hospitals and health centers have taken it upon themselves to obtain the data directly from people with a known diagnosis or comorbidity, using the appropriate technique to obtain it, and then recording these results. With related clinical information, profiles with conclusive diagnosis in cases of T2DM and observable factors can be taken and be compared with other clinical profiles of people without diabetes prognosis, but with risk factors (control group). Some risk factors are: overweight, high Low Density Lipoproteins (LDL), cholesterol or irregular systolic blood pressure (SBP), among others [8]. These people should be the initial patients to be compared, as there is a high probability that they are at least prediabetic or may become part of the T2DM risk group in the near future. As the diagnosis and treatment of T2DM takes time and is expensive for these institutions, it is necessary to use specific data to develop low-cost models that help with the prognosis and save time and costs in the process, reducing the progress and complications of a possible illness.

Machine Learning (ML) techniques have been used in recent years to use this data and approach to solve or at least predict different diagnostic problems, such as diabetes [7], COVID-19 [9], heart disease [10], Alzheimer’s [11], among others. An ML model can use certain data given a specific set of parameters and return a prediction as a result, which can be later tested or checked against another set of data or untrained data from the same database to validate the model, using the Area Under the Curve (AUC) among other performance metrics to identify the variation or importance of a certain feature or group of features in the detection of T2DM. The ensemble models are very important as they consist in group or stack different models in order to have more robust results confirming or complementing the final result of the experiments. The ensemble approach such as the work performed by Cohen S. et al., (2021) [12], or contributing in the performance with improved models [13] or treatment of bias [14], provide tools to group and improve models to generate better or more robust results. The use of ML models to support the treatment or monitoring of the physical state of a patient with T2DM presents important advantages, one of the most notable is the reduction of continuous invasive glucose monitoring in insulin dependent people. In the approximation from Daniel N. Thyde et al., (2021) who proposed detecting adherence to once-daily basal insulin injections by machine learning, this approach using logistic regression and convolutional neural networks; the results of this product obtained an average accuracy of 79.7% [15]. This contribution provides a guideline in the sense that it is necessary to make use of this type of tools in order to reduce the invasive and repeated check of traditional glucometers and focus on the variations obtained by continuous glucose monitoring, all this verified in simulated environments (the medtronic virtual patient software is used). Hasan K. et al., (2020) propose an ensemble by weight for diabetes prediction with the use of different models of machine learning: k-nearest Neighbour, Decision Trees, Random Forest, AdaBoost, Naive Bayes, and XGBoost; does data standardization with Z-Score, Principal Component Analysis-based dimensionality reduction, 5-fold cross-validation, and Multilayer Perceptron (MLP) were employed. The weighted ensembling of different ML models is also proposed, to improve the prediction of diabetes where the weights are estimated from the corresponding Area Under ROC Curve (AUC) of 0.950 (95%) [16]. This ensemble uses the PIMA Indian dataset [17]. The proposal of Syed A.H. et al., (2020) is to develop a questionnaire-based cross-sectional study using conventional diabetes risk factors to study prevalence and association between outcomes and exposure. The best performing classifier’s hyper-parameters were further tuned with a 10-fold cross-validation and F1 score to select the final model. The purpose of the study is to estimate the prevalence of the disease and calculate the odds ratio to measure the association between exposure (explanatory variable) and the outcome variable in the questionnaire based on the research plan. A decision tree model is proposed with a final AUC of 0.88 (88%) [7]. Another use of ML models in diabetes, is reflected in the study of Kazuya Fujihara et al., (2021) who propose to develop machine learning models for decision making on the initiation of insulin treatment in Japanese patients with T2DM, their purpose is to examine the ability of machine learning models to predict the initiation of insulin use given by specialists and whether the machine learning approach could support decision making by general practitioners for insulin initiation in patients with T2DM. The models used were logistic regression and neural networks and the results were an AUC of 0.89–0.90 (89–90%) for logistic regression and 0.67–0.74 (67–74%) for ANN [18]. The use of ensemble by stacking and soft voting proposed by Deberneh H. and Kim I. (2021) provides the use of random forest, support vector machine and XGBoost as ML models to predict if a patient is non-diabetic, pre-diabetic or diabetic. The features analyzed in this study have similarities to the experiment presented, some of these features are: plasma glucose (FPG), Glycated hemoglobin(HbA1c), triglycerides, BMI, age, uric acid, sex. The results obtained contains 12 features to construct the models and an accuracy of the ensemble in 78% [19]. Kocbek S. et al., (2022) implements the Least Absolute Shrinkage and Selection Operator (LASSO) for feature selection, with 80–20% split data for training and testing, respectively. To predict undiagnosed T2DM, a Logistic Regression model is used, incorporating features such as: glucose level (FPGL), age, gender, BMI and Waist circumference with the AUC of 0.818 (81.8%) as a single model proving that the LASSO-Logistic Regression combination is a potent prediction tool to help as “interpretable models in healthcare that can contribute to higher trust in prediction models from healthcare experts” [20]. Shaker E. et al., (2019) [21] proposes a framework that classifies Diabetes by combining (ensemble) k-nearest neighbors, naïve Bayes, decision tree, support vector machine, fuzzy decision tree, artificial neural network, and logistic regression and Is evaluated using a real dataset collected from electronic health records of Mansura University Hospitals (Mansura, Egypt). The results achieved are 90% of accuracy, 90.2% of recall = 90.2%, and 94.9% of precision. Kumari et al., (2021) [22] proposed ensemble soft voting classifier gives binary classification and uses the ensemble of three machine learning algorithms viz. random forest, logistic regression, and Naive Bayes for the classification. Empirical evaluation of the proposed methodology has been conducted with state-of-the-art methodologies and base classifiers such as AdaBoost, Logistic Regression, Support Vector machine, Random forest, Naïve Bayes, Bagging, GradientBoost, XGBoost, CatBoost. by taking accuracy, precision, recall, F1-score as the evaluation criteria. The proposed ensemble approach gives the highest accuracy, precision, recall, and F1-score value with 79.04%, 73.48%, 71.45% and 80.6% respectively on the PIMA diabetes dataset. Further, the efficiency of the proposed methodology has also been compared and analysed with breast cancer dataset. The proposed ensemble soft voting classifier has given 97.02% accuracy on the breast cancer dataset. Singh N. et al., (2020) [23] develop a stacking-based evolutionary ensemble learning system “NSGA-II-Stacking” for predicting the onset of Type-2 diabetes mellitus (T2DM) within five years. For this purpose, publicly accessible Pima Indian diabetes (PID) dataset is utilized. As a data pre-processing step, the missing values and outliers are identified and imputed with the median values. For base learner selection, a multi-objective optimization algorithm is utilized which simultaneously maximizes the classification accuracy and minimizes the ensemble complexity. As for model combination, K-nearest neighbor (K-NN) is employed as a meta-classifier that combines the predictions of the base learners. The comparative results demonstrate that the proposed NSGA-II-Stacking method significantly outperforms several individual ML approaches and conventional ensemble approaches. In terms of performance metrics, the proposed system achieves the highest accuracy of 83.8%, sensitivity of 96.1%, specificity of 79.9%, f-measure of 88.5% and area under ROC curve of 85.9%. Liu Y. et al., (2019) [24] proposed the use of machine learning algorithms to improve the accuracy of type 2 diabetes predictions using non-invasive risk score systems. The least absolute shrinkage and selection operator (LASSO), smoothed clipped absolute deviation (SCAD), and minimax concave penalized likelihood (MCP), which are commonly used on selecting variables for high-dimensional models, were used to automatically select significant non-invasive risk factors for Type 2 diabetes. A more conservative model selection method for ultrahigh-dimensional model, the iterative sure independence screening (ISIS) procedure for variable selection in logistic regression and the traditional stepwise logistic regression were also applied to this dataset. Support vector machine, tree-based methods (e.g., decision tree and random forest), and neural network were three commonly used machine learning techniques for diabetes risk prediction, the results of the ensemble model with a feature selector are AUC of 0.802 (0.780, 0.825), Sensitivity of 0.662 (0.614, 0.709) and Specificity of 0.702 (0.676, 0.728). All the studies presented as related work, have developed models with a precision based mainly on Glucose levels (Obtained by fasting plasma glucose (FPG) or oral glucose tolerance test (OGTT)) or the H1A1c and selecting features as complement by making a comparison between several others using stacking, ensemble or reflecting the results of each model individually, it can be asserted that: Dependence on glucose levels or Hb1Ac to diagnose T2DM is a common practice but is not conclusive and strengthening the results of individually validated models with other features is mandatory; for this, the ensemble of models integrates the results of each model used in a single set, identifying the best possible scenario with a selection of features defined by a supported technique, and with the omission of glucose levels and HbA1c as part of the models implemented, there is a possibility of identifying other features that can be considered individually or grouped, to have the potential to become biomarkers of T2DM so that they can be tested, replicated and used in diagnosis by the medical area. The proposal in this study is to implement an ensemble ML model to classify patients with and without T2DM, based on clinical data. Using these data as a non-invasive, practical, extremely rapid and accurate approach similar to FPG, OGTT or HbA1c could be given. This model seeks to have exclusively features non-related with glucose. With the use of multivariate models, which is, the use of multiple features (Non-invasive and non-related glucose features in this work) applied into a model so correlations or patterns can be found, used for the detection of T2DM. The proposed ML ensemble model focuses on classifying the 1787 patients with an input of the 48 availables in the processed database. The dataset for this study is acquired from “Unidad de Investigación Médica en Bioquímica, Centro Médico Nacional Siglo XXI, IMSS”, with the information of Mexican patients. The database contains anthropometric data, medical treatment, complications, lipid profile and blood pressure. It is worth mentioning that there are no public datasets with this type of patients.

This work is divided into 4 sections, Section 1 is this introductory part, Section 2 describes the data, models and the methodology used to carry out the development of the ensemble model and how it is validated. Section 3 shows the results obtained and a detailed analysis is included using the output graphs. Finally, Section 4 shows discussions, conclusion and future work.

## 2. Materials and Methods

The methodology in Figure 1 is based on the metodology proposed by Akhtar T. et al., (2021) [25] and is explained as follows: the first step is to carry out a preliminary analysis of the sample data, followed by data processing and imputation, preparing the folds and data separation in test and training, performing the selection of features, developing the models and integrating them into the ensemble to finally, validate the models with the test data and extract the AUC, sensibility and specificity.

### 2.1. Sample

The database was provided by the Centro Médico Nacional Siglo XXI located in Mexico City. All Mexican patients signed an informed consent letter and the protocol meets the Helsinki criteria which were approved by the Ethics Committee of Instituto Mexicano del Seguro Social under the number R-2011-785-018. It includes 1787 patients, 898 positive cases of T2DM and 889 control patients, according to gender, this database includes: 892 men and 895 women. In Figure 2 shows the Starting correlation between the features.

### 2.2. Data Treatment

This step of the process includes the treatment of data, beginning with data imputation, focusing on missing values and features outside the scope of this study; the next step is the data imputation values included in the database and the final step is the normalization of the data.

### 2.3. Data Imputation

This work uses an exclusion criteria of working only with the observations that have complete data for all variables and discarding all others, removing all out-of-scope or unusable features [26], the features that identify patients, and IDs were removed, since they were only internal identification data of the patient and a consecutive number, respectively. As in the dataset exists a series of features and patients with missing data, all null or not available data (NA) were removed, since the features identified with NA were exclusive to the cases (patients positive for T2DM) and show nonexistent data in the control patients (patients negative for T2DM) or these data were not collected or recorded, therefore, they were not taken into consideration. Based on this criteria, all medications and their daily intake amount were eliminated: Glibenclamide, Glibenclamide dose, Metformin, Metformin dose, Pioglitazone, Pioglitazone dose, Rosiglitazone, Rosiglitazone dose, Acarbose, Acarbose dose, Insulin and Insulin dose. Data from 3 patients, these patients (in all 3 cases) had no data on the feature Hypertension under treatment; the features Glomerular filtration rate, Age of diagnosis (age at which the positive diagnosis of T2DM is presented, data exclusively in cases), HbA1c and T2DM complications were also eliminated, since there are only present data for patients with T2DM. The T2DM complications feature contains comorbidities associated with T2DM and are also outside the scope of this experiment, as it contains NA in some of the cases and NA in all control patients.

Since the Glucose feature is a well-known biomarker, it is removed from the scope of this experiment in order to review the performance of the other features. This feature in univariate models test had over 90% AUC making it the most important of the features on the dataset analyzed.

All features eliminated are listed in Table 1 and the features analyzed are listed in Table 2 that corresponds to all features included in LASSO and the models.

### 2.4. Normalization of Data

Since most features in the dataset analyzed have a different range of values, Z-Score is implemented so the features can be compared to one another or be taken as part of the final model more accurately setting them in the same range. The Z-Score consists of: calculating the mean of the values in a feature, calculating the standard deviation of the same values and finally calculating the Z-Score with the following formula [27]:(1)$$Z=\frac{x-\bar{x}}{\sigma_{x}},$$
where *x* is the raw score, *x* the mean of all the *x* and σ the standard deviation of *x*. This process results in a value in a standard range, which is applied to different features. A generic function whose default method centers and/or scales the features of a numeric matrix has been implemented.

### 2.5. Feature Selection

LASSO is used in this work as it has the minor *P*-value of the feature selectors proposed by Liu Y. et al., (2019) [24], the other selectors are: smoothed clipped absolute deviation (SCAD), minimax concave penalized likelihood (MCP), stepwise logistic regression and the iterative sure independence screening (ISIS). The dataset is under 10,000 patients and is under the inclusion criteria of this work as it has several similarities in data set size and features contained, including the T2DM classification (if the patient has T2DM or not), body mass index (BMI),diastolic blood pressure (DBP), age, sex, Waist Circumference (considered a better indicator than the WHR contained in this work), systolic blood pressure (SBP), high density lipids (HDL), low density lipids (LDL), cholesterol, and other features related to glucose. From 22 features presented in Table 2 after the preprocessing and elimination of non-relevant or non-related to glucose features for the inclusion in the selection process, LASSO is implemented, as a fast and solid solution as is presented in the study conducted by Kocbek S. et al., (2022) [20], were sets an adjusted to a generalized linear models and similar through the penalized maximum likelihood. The developed LASSO implementation solves the problem:(2)$$\begin{array}{c} \min\\ \left(\beta_{0},\beta \right)\in R^{p+1} \end{array}\left[\frac{1}{2N}\sum_{i=1}^{N}{\left(y_{i}-\beta_{0}-x_{i}^{T}\beta \right)}^{2}+\lambda P_{a}\left(\beta \right)\right],$$
where
(3)$$P_{a}\left(\beta \right)=\left(1-\alpha \right)\frac{1}{2}{∥\beta ∥}_{\ell_{2}}^{2}+\alpha {∥\beta ∥}_{\ell_{1}}$$
(4)$$=\sum_{j=1}^{p}\left[\frac{1}{2}\left(1-\alpha \right)\beta_{j}^{2}+\alpha \left|\beta_{j}\right|\right]$$
is the elastic-net penalty as Zou and Hastie (2005) exposes [28]. $P_{\alpha }$ is a compromise between the ridge-regression penalty (α=0) and the LASSO penalty (α=1). $\beta_{0}$ is the constant coefficient and β a vector of coefficients. The elastic-net penalty is controlled by α and bridges the gap between the LASSO regression (α = 1, the default value) and the Ridge regression (α = 0). The tuning parameter λ controls the overall strength of the penalty. This implementation used a cross-validation approach, which performed a k-fold cross-validation for glmnet and returns a value for λ [29].

The ridge penalty is known to reduce the coefficients of correlated predictors as they get closer to each other, while LASSO tends to choose one correlated coefficient and discard the others. The elastic-net penalty performed by the glmnet package [29] combines these two: if the predictors are correlated in groups, α = 0.5 tends to select or leave out the entire group of features, resulting in a set of usable hyperparameters for the model.

### 2.6. Development of Ensemble Model Tests

Once the data treatment is complete and the features are selected, the development of the ensemble model with data partitioning begin, this process is to ensure that the results of the training and tests are as accurate as possible, avoiding overfitting or mismatching. In order to obtain the best ensemble model by hard voting, each model is validated with AUC starting with the development of the Generalized Linear Regression model (GLM), then the Support Vector Machine model (SVM), then an Artificial Neural Network with a single hidden layer is implemented and finally, the comparisons are made to carry out the majority vote process, for the final integration of the ensemble model.

### 2.7. Data Partition

For data partitioning, it is considered to divide the database into 75% for training and 25% for tests using cross validation. The goal of this split is to get an index of each row to separate the data in this proportion and then assign it to a dataset (partition) and for the configuration of the parameters, a 10-fold is established, as is used by Syed A.H. et al., (2020) [7] in biomedical applications. The percentage of 75% and 25% obtained after the comparisons in the iterations, establishing the best scenario with the models used in the ensemble implemented, the 10-fold cross-validation is enough to balance the training and testing results correctly, giving the best possible AUC results for all three models.

### 2.8. Ensemble Model

To do the ensemble, the Hard voting method is implemented, which gives us a robust result as the one presented by Deberneh H. and Kim I. (2021) [19] to predict if a patient is non-diabetic or diabetic, although it does not considerably reduce the classification error of the predictions when comparing the AUC result between each model, reaffirms the result repeatedly.

The 3 models implemented for the ensemble are: Generalized Linear Model (GLM), Support Vector Machines (SVM) and Artificial Neural Networks (ANN) as shown in this research performed by Kavakiotis I. et al. [6]. These models were selected because they are well known for making accurate predictions in classification problems and broadly used in clinical data processing; all of these models used the product features of the LASSO implementation.

This model by specifications does not require hyperparameters tuning.

### 2.9. Generalized Linear Model

The generalized model used is a Logistic Regression (LR), a specific model is built, giving a symbolic description of the linear predictor and a description of the error distribution. Consider the systematic effects of the linear model with:(5)$$S\left(t\right)=\frac{1}{\left(1+e^{-t}\right)}$$
the sigmoid function described (*S*), keeps the value of within the [0,1] range. It searches a value for the dichotomic classification of 1 when the probability is larger or closer to 1, and 0 when *x* is smaller [28].

The hyperparameters obtained in LASSO (shown in Table 3), were integrated in this model as part of this experiment.

This model by specifications does not require hyperparameters tuning.

### 2.10. Support Vector Machines

The radial kernel used in the support vector machine model in this ensemble works similarly to KNN (K-Nearest Neighbors) by fitting the closest observations into the new observation, grouping them based on how much they influence the output of the set. classifier for multiple hyperplanes.

The radial kernel is represented by:(6)$$K\left(x,x^{\prime }\right)=e^{\frac{-||x-x^{\prime }{\left|\right|}^{2}}{2\sigma^{2}}},$$
where *x* and $x^{\prime }$ are original observations and new observations, respectively [30].

The hyperparameters obtained in LASSO (shown in Table 3), were integrated in this model as part of this experiment.

The configuration of the hyperparameters in this model is: Cost C=1, Gamma λ=0.2 and the Kernel=Radial.

### 2.11. Artificial Neural Network

For the ANN model it is fitted to a single hidden layer, possibly with skip layer connections and the problem is solved:(7)$$f\left(net\right)=\sum_{j=1}^{n}\left(w_{j}x_{j}\right),$$
donde *x* is a node of the neural network and *w* represents the weight of each network layer that connects one node to another [31].

The hyperparameters obtained in LASSO (shown in Table 3), were integrated in this model as part of this experiment. The configuration of the hyperparameters in this model is: weights=1, size=1 and entrophy=least-squares.

### 2.12. Implementation

All models and methodology are implemented in R, a well-known open source software validated by the scientific community, as well as the following libraries:LASSO is implemented using “glmnet” [29].The Generalized Linear Model is implemented with “caret” [32].For Support Vector Machines “caret” [32] is used.The Artificial Neural Network is implemented in R with “caret” [32].The ensemble is implemented in R with nested decisions (use of if).

## 3. Results

The data analyzed consisted in 1787 patients, 898 positive cases of T2DM and 889 control patients all of them with the features described in Table 1 and Table 2. The data treatment consisted in data imputation (Section 2.3) and normalization of data (Section 2.4), resulted in a balanced and normalized data and 10 features discarded (Table 1). The 23 features processed in LASSO (Section 2.5) with elastic-net penalty obtained a set of 12 features for the models integrated in the ensemble model, these features are shown in Table 3.

All the results in Tables 6, 8, 10 and 12 were calculated (8)$$Sensitivity=\frac{T_{p}}{\left(T_{p}+F_{n}\right)},$$
(9)$$Specificity=\frac{T_{n}}{\left(F_{p}+T_{n}\right)},$$
(10)$$Precision=\frac{T_{p}}{\left(T_{p}+F_{p}\right)},$$
(11)$$NegativePredictiveValue=\frac{T_{n}}{\left(T_{n}+F_{n}\right)},$$
(12)$$FalsePositiveRate=\frac{F_{p}}{\left(F_{p}+T_{n}\right)},$$
(13)$$FalseNegativeRate=\frac{F_{n}}{\left(F_{n}+T_{p}\right)},$$
(14)$$Accuracy=\frac{\left(T_{p}+T_{n}\right)}{\left(T_{p}+T_{n}+F_{p}+F_{n}\right)},$$
(15)$$F1Score=\frac{2T_{p}}{\left(2T_{p}+F_{p}+F_{n}\right)}$$
where: $T_{p}=$ True positive, number of subjects with T2DM correctly classified.$F_{p}=$ False positive, number of healthy subjects incorrectly classified.$T_{n}=$ True negative, number of healthy subjects correctly classified.$F_{n}=$ False negative, number of subjects with T2DM classified as healthy.

These metrics provide which of the models implemented is best for identifying T2DM patients, in Table 4 shows the description of each metric and in Table 5 shows the structure used to calculate each element presented in the confusion matrix.

The models developed showed good performance in the implementation of the ensemble, the SVM model with a radial kernel had an AUC of 92.8% ± 3% with the 25% test set, in the confusion matrix of the SVM model, is observed that the sensitivity of 0.8750 (87.5%) and is lower than the specificity of 0.9238 (92.38%) as presented in Table 6 and the confusion matrix in Table 7.

The ANN model with a single layer had an AUC of 90.5% ± 3% with the 25% test set, and the confusion matrix of the ANN model, is observed that the sensitivity of 0.8559 (85.59%) is lower than the specificity of 0.9175 (91.75%), this second model shows the lower sensitivity and specificity compared to the SVM model as shown in Table 8 and the confusion matrix in Table 9.

The GLM model had an AUC of 90.5% ± 3% with the 25% test set, in the confusion matrix of the GLM model, is observed that the sensitivity of 0.8487 (84.87%) is lower than the specificity of 0.9167 (91.67%) as it is shown in Table 10 and the confusion matrix shown in Table 11. This model presets the lowest sensitivity and specificity of all models including the ensemble.

The ensemble model with max voting had AUC of 90.5% ± 3%, this AUC shows consistency with the 3 models integrating a robust solution to the classification problem. The confusion matrix of the ensemble model, observed that the sensitivity of 0.8788 (87.88%) is lower than the specificity of 0.9242 (92.42%) shown in Table 12 and the confusion matrix in Table 13, values above average the sensitivity and specificity of the models implemented.

The ensemble model showed an AUC similar to the ANN and GLM models with the 90.5% as average, with the SVM model being 2.3% better than the other models in this case, as shown in Figure 3.

## 4. Discussion

The proposed methodology shows potential to solve a dichotomic classification problem and it can be determined that the features included in the experiments conducted are directly related for the detection of T2DM, as it is presented in the results. The data partition of 75% for training 25% for blind testing with a 10-fold cross-validation [33] is sufficient to provide consistency in the implementation of LASSO as a feature selection method and all the models implemented, resulted in an AUC in all the implementations proofs consistency along all data analyzed, with the set of 12 features product as hyperparameters used to train and test the models, the features produced by this methodology are:

**Salary**: With the data analyzed, it can be deduced that people with incomes greater than 5000 Mexican pesos tend to have healthier anthropometric data, lower lipid levels and better blood pressure, having less risk of suffering from T2DM than people with incomes less than 5000 Mexican pesos. This relates statistically in Latin America (low- and middle-income countries specifically) as the low and middle income class is more likely to be in risk of T2DM [34]. From the data of the feature **Sex**, it is implied that male patients classified as positive are more than female, achieving a direct relationship with the statistics shown in the introduction and the sample, yet this consistency does not present significant interaction with the other features as presents Gou W. et al. in the construction of a microbiome with similar features comparing it with the sex feature [35]. The feature **Age** values with the highest risk of developing T2DM according to the analyzed data, is between 44 and 58 years old. **WHR** and **BMI** derived from the data show that these features are directly related to obesity [36], if the values are in between the risk factor of obesity classes as it shows in the classification of obesity given by Center for Disease Control and Prevention (CDC) [37], these cases will increase the risk to develop the disease of T2DM as the age and the BMI increases [38]. The **Urea** levels are directly associated with an increased risk of T2DM, another feature with potential use for screening, and presented as key factor in detection of diabetes by Dinh A. et al. [39]. **Lipids in treatment** are somehow related to T2DM, as it’s variability can be used in diabetes monitoring as supports Lee S. et al. [40], but this feature can be discarded in this experiment as untreated lipid levels and type of medication for treatment are not shown in the data, making it impossible to make comparisons and identify differences or variances, it only shows whether or not medication was used to correct the levels. The **HDL** feature is a significant predictor for T2DM as it confirms Lai H. et al. [41] showing direct correlation with **Triglycerides**, SBP and **BMI**. The **Hypertension under treatment**, **DBP** or Diastolic Blood Pressure and **uncorrected SBP** or medication uncorrected Systolic Blood Pressure are highly correlated with T2DM. In the case of this experiment the different factor is the correction of levels through medication in the hypertension, this correlation was reviewed and compared with no significant difference with the study of Zheng H. et al. [42] where the blood pressure and hypertension weren’t controlled or corrected by medication. Diastolic Blood Pressure is the most relevant feature in the 3 models implemented in this experiment and is a potential biomarker for the detection of T2DM worthy of future studies.

As main findings it can be established that a model without glucose related features have similar performance as the models with glucose related features as it is shown in the Table 14:

The false positive and false negative rates of the ensemble proposed, clearly presents excellent results, as shown in the Table 12, this low rates explain that the model is solid and usable to support accurate diagnostics in real world situations as the scenarios presented in every patient data analyzed.

The AUC of the SVM is slightly better that the rest of the models including the final ensemble, nether less, this could change as the random nature of the partition sets changed, performing worst in some cases, the scenario presented in this work is included as it provided an outcome achieving over 90% AUC.

## 5. Conclusions

The results obtained were 90% ± 3% in the GLM–SVM–ANN ensemble model using LASSO for feature selection and dividing the data into 75% for training and 25% for testing, validated by 10-fold cross-validation and AUC are satisfactory but conclusive. The percentage of AUC is close to the goal of 95% to be used in a clinical setting, the need to implement new ways of processing the data and still avoid the Glucose feature or use it only as a reference, as proposed, ensures obtaining features with predictive potential.

As a disadvantage, inconsistencies were obtained in some LASSO runs, in this case, throwing a thirteenth feature that varied alternating between: Education, uncorrected LDL and SBP, these features were shown by the randomness product of cross-validation and were not taken as part of the final model. For this reason, otherwise, the features that were selected persisted in each run even with the change of data used in the different samplings that each separation generated by the folds performed.

All models confusion matrices including the final ensemble obtained a 3% to 4% sensitivity higher than the specificity, this behavior is considered within the accepted parameters for the detection of a disease.

The machine learning algorithms and the clinical data used for this methodology, showed potential to identify relationships, predictions and behavior patterns that classify over 90% of the patients accurately with or without T2DM, using biomarkers extracted by non-invasive methods with the same or superior precision than the invasive laboratory blood tests: FPG, OGTT or the H1A1c.

## 6. Future Work

For future research, it is proposed to delve into the relationship between Diastolic Blood Pressure and T2DM. Another proposal is to use a different type of imputation to have a different approach with some of the discarded features, using other feature selection approaches like genetic algorithms or Boruta and trying new models, hoping to successfully reach or exceed 95% AUC and ultimately discover potential biomarkers.

## Figures and Tables

**Figure 1 healthcare-10-01362-f001:**
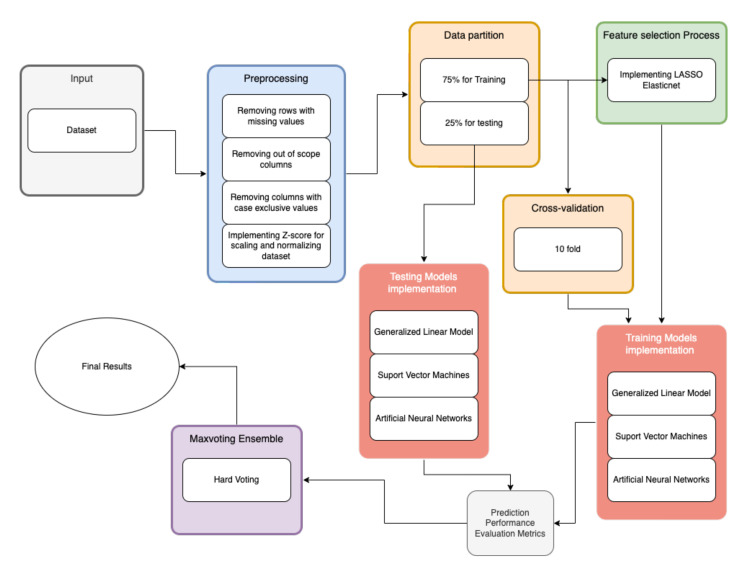
Flowchart of the proposed methodology.

**Figure 2 healthcare-10-01362-f002:**
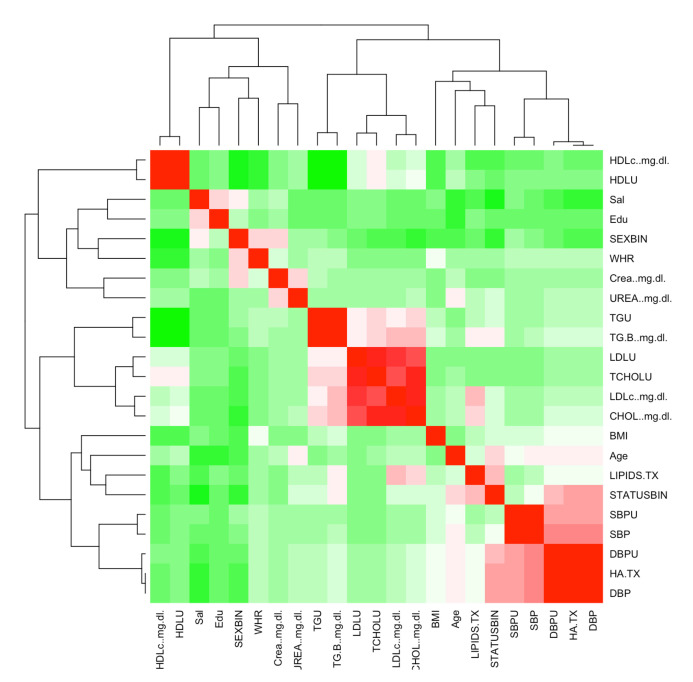
Feature Correlation Heat Map.

**Figure 3 healthcare-10-01362-f003:**
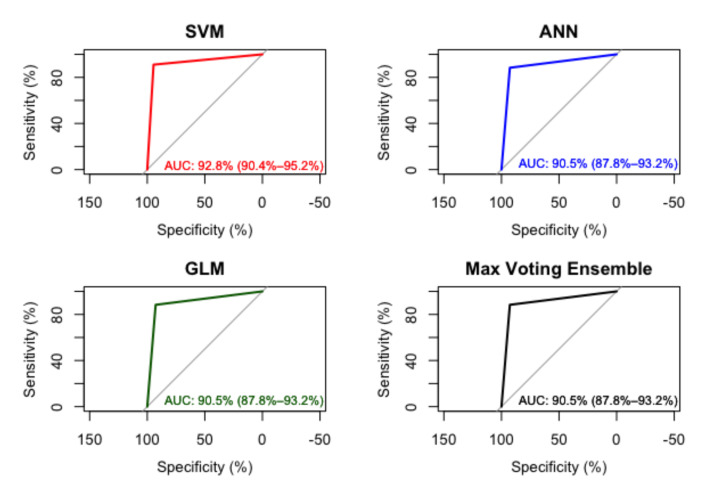
AUC in SVM, ANN, GLM and Ensemble Model.

**Table 1 healthcare-10-01362-t001:** Features discarded.

Feature	Description	Possible Values
Age DX	Diagnosis age of T2DM	Numeric Integer
Glucose	Blood glucose levels	Numeric
HbA1c	Glycated Hemoglobin	Numeric
GFR	Glomerular Filtration Rate (blood test that checks how well the kidneys are working)	Numeric Integer
Glibenclamide	Drug Treatment	0—No 1—Yes
Metformin	Drug Treatment	0—No 1—Yes
Pioglitazone	Drug Treatment	0—No 1—Yes
Rosiglitazone	Drug Treatment	0—No 1—Yes
Acarbose	Drug Treatment	0—No 1—Yes
Insuline	Drug Treatment	0—No 1—Yes
Complications T2DM	Complications associated with T2DM	NEUROPATHY—Have neuropathy RETINOPATHY—Have retinopathy

All features in this table were excluded from the analysis by data imputation.

**Table 2 healthcare-10-01362-t002:** Features description and possible values.

Feature	Description	Possible Values	*p*-Value Univariate Logistic Regression
Education	Studies concluded by the patient	1—Elementary School 2—Secondary School 3—Technical level 4—High School 5—Professional 6—Postgraduate	0.00118
Salary	Monthly income	1—Less than $2000.00 2—Between $2000.00 and $5000.00 3—More than $5000.00	$2^{-16}$
Sex	Patients sex	0—Male 1—Female	$6.7^{-16}$
Age	Age of the patient in years	Numeric Integer	$2^{-16}$
WHR	Waist Hip Ratio	Numeric	$2.89^{-5}$
BMI	Body Mass Index	Numeric	$9.35^{-16}$
Urea	Waste product resulting from the breakdown of protein in the patient body	Numeric Integer	$2^{-16}$
Creatinine	Waste product produced by muscles as part of regular daily activity	Numeric	0.000456
Lipids treatment	Lipid levels in treatment	1—Lipid levels in treatment 2—Lipid levels without treatment.	0.956
Cholesterol	Fat-like substance that is found in all cells of the patient body	Numeric	$2^{-16}$
HDL	High Density Lipoprotein (corrected for medication)	Numeric	$3.77^{-12}$
LDL	Low Density Lipoprotein (corrected for medication)	Numeric	$2^{-16}$
Triglycerides	Type of fat found in the patient body	Numeric	$2^{-16}$
TCHOLU	Total Cholesterol (uncorrected)	Numeric Integer	0.258
HDLU	High Density Lipoprotein (uncorrected)	Numeric Integer	$2.12^{-5}$
LDLU	Low Density Lipoprotein (uncorrected)	Numeric Integer	0.240
TGU	Triglycerides (uncorrected)	Numeric Integer	$2^{-16}$
SBP	Systolic Blood Pressure (corrected by medication)	Numeric Integer	$2^{-16}$
DBP	Diastolic Blood Pressure (corrected by medication)	Numeric Integer	$2^{-16}$
SBPU	Systolic Blood Pressure (uncorrected)	Numeric Integer	$1.28^{-10}$
DBPU	Diastolic Blood Pressure (uncorrected)	Numeric Integer	$2^{-16}$
HA-TX	Hypertension Treatment	0—Not in hypertension treatment 1—In hypertension treatment	0.959
Output	Classifier of patients	0—Patient negative for T2DM 1—Patient positive for T2DM	-

All features in this table were included in the analysis by data imputation.

**Table 3 healthcare-10-01362-t003:** LASSO result Features.

Feature	Description	Possible Values	*p*-Value Multivariate LASSO Selection Logistic Regression
Salary	Monthly income	1—Less than $2000.00 2—Between $2000.00 and $5000.00 3—More than $5000.00	$2.72^{-16}$
Sex	Patients sex	0—Male 1—Female	0.00538
Age	Age of the patient in years	Numeric Integer	$3.05^{-13}$
WHR	Waist Hip Ratio	Numeric	0.07312
BMI	Body Mass Index	Numeric	0.00760
Urea	Waste product resulting from the breakdown of protein in the patient body	Numeric Integer	$7.43^{-7}$
Lipids treatment	Lipid levels in treatment	1—Lipid levels in treatment 2—Lipid levels without treatment	0.97047
HDL	High Density Lipoprotein (corrected by medication)	Numeric	$1.19^{-7}$
Triglycerides	Type of fat found in the patient body	Numeric	$6.31^{-5}$
DBP	Diastolic Blood Pressure (corrected by medication)	Numeric Integer	$2^{-16}$
SBPU	Systolic Blood Pressure (uncorrected)	Numeric Integer	$3.56^{-5}$
HA-TX	Hypertension Treatment	0—No 1—Yes	0.96440

All features in this table were included in all models as part of the final ensemble.

**Table 4 healthcare-10-01362-t004:** Metrics.

Metric	Description
Sensitivity (see Equation (Equation 8))	Correct identification of patients with T2DM (True Positive)
Specificity (see Equation (Equation 9))	Correct identification of patients without T2DM (True Negative)
Precision (see Equation (Equation 10))	Defines what portion of the positive cases of T2DM are actually positive
Negative Predictive Value (see Equation (Equation 11))	Defines what portion of the negative cases of T2DM are actually negative
False Positive Rate (see Equation (Equation 12))	The rate of the predicted false values that are actually true
False Negative Rate (see Equation (Equation 13))	The rate of the predicted true values that are actually false
Accuracy (see Equation (Equation 14))	The percentage of cases that the model has classified correctly
F1 Score (see Equation (Equation 15))	The measure of precision that a test has

All metrics in this table were extracted in models as part of the final ensemble.

**Table 5 healthcare-10-01362-t005:** Confusion Matrix structure.

True Values	Predicted (True)	Predicted (False)
True	$T_{p}$	$T_{n}$
False	$F_{p}$	$F_{n}$

**Table 6 healthcare-10-01362-t006:** SVM Confusion Matrix Measure Values.

Measure	Value
Sensitivity	0.8750
Specificity	0.9238
Precision	0.9269
Negative Predictive Value	0.8700
False Positive Rate	0.0762
False Negative Rate	0.1250
Accuracy	0.8982
F1 Score	0.9002

**Table 7 healthcare-10-01362-t007:** SVM Confusion Matrix.

True Values	Predicted (True)	Predicted (False)
True	203	16
False	29	194

**Table 8 healthcare-10-01362-t008:** ANN Confusion Matrix Measure Values.

Measure	Value
Sensitivity	0.8559
Specificity	0.9175
Precision	0.9224
Negative Predictive Value	0.8475
False Positive Rate	0.0825
False Negative Rate	0.1441
Accuracy	0.8846
F1 Score	0.8879

**Table 9 healthcare-10-01362-t009:** ANN Confusion Matrix.

True Values	Predicted (True)	Predicted (False)
True	202	17
False	34	189

**Table 10 healthcare-10-01362-t010:** GLM Confusion Matrix Measure Values.

Measure	Value
Sensitivity	0.8487
Specificity	0.9167
Precision	0.9224
Negative Predictive Value	0.8386
False Positive Rate	0.0833
False Negative Rate	0.1513
Accuracy	0.8801
F1 Score	0.8840

**Table 11 healthcare-10-01362-t011:** GLM Confusion Matrix.

True Values	Predicted (True)	Predicted (False)
True	202	17
False	36	187

**Table 12 healthcare-10-01362-t012:** Maxvoting Ensemble Confusion Matrix Measure Values.

Measure	Value
Sensitivity	0.8788
Specificity	0.9242
Precision	0.9269
Negative Predictive Value	0.8744
False Positive Rate	0.0758
False Negative Rate	0.1212
Accuracy	0.9005
F1 Score	0.9022

**Table 13 healthcare-10-01362-t013:** Ensemble Model Confusion Matrix.

True Values	Predicted (True)	Predicted (False)
True	203	16
False	28	195

**Table 14 healthcare-10-01362-t014:** Related work comparison.

Autor	ML Model	Dataset	Metrics
Shaker E. et al., (2019) [21]	Ensemble of: k-nearest neighbors, naïve Bayes, decision tree, support vector machine, fuzzy decision tree, artificial neural network, and logistic regression	Electronic health records of Mansura University Hospitals (Mansura, Egypt)	90% of accuracy, 90.2% of recall, and 94.9% of precision
Kumari et al., (2021) [22]	Ensemble of: random forest, logistic regression, and Naive Bayes	PIMA diabetes dataset	79.04% of accuracy, 73.48% of precision, 71.45% of recall, and 80.6% of F1_score
Singh N. et al., (2020) [23]	stacking-based evolutionary ensemble learning system “NSGA-II-Stacking”	PIMA diabetes dataset	accuracy of 83.8%, sensitivity of 96.1%, specificity of 79.9%, f-measure of 88.5% and area under ROC curve of 85.9%
Liu Y. et al., (2019) [24]	Majority voting Ensemble: Support vector machine, tree-based methods and neural networks	REACTION study (Risk Evaluation of Cancers in Chinese Diabetic Individuals: A Longitudinal Study)	Majority voting with model selection results: AUC of 0.802 (80.2%), Sensitivity of 0.662 (66.2%), Specificity of 0.702 (70.2%)
This Work Authors	Hard voting Ensemble of: generalized liner regression, support vector machines and artificial neural networks	Centro Médico Nacional Siglo XXI dataset	Sensitivity of 0.8788 (87.88%) Specificity of 0.9242 (92.42%) Precision of 0.9269 (92.69%) Area under the ROC curve 90.5%

## Data Availability

Not applicable.

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
