# Peer review of "Hard Voting Ensemble Approach for the Detection of Type 2 Diabetes in Mexican Population with Non-Glucose Related Features"

_healthcare, 2022, doi:10.3390/healthcare10081362_

Round 1
Reviewer 1 Report
I have following comments before proceeding further:
1. Figure 1 resolution is very less
2. Explain novelty of your work
3. What you mean non dichotomous data, kindly explain in your paper.
4. What you infer in Z score from your work, please include.
5. How you used tuning parameter as variable parameters.
6. Which kernel are used in SVM.
7. Kindly compare your results with minimum three recent papers
Author Response
List of changes
The paper reviewed have text color changes so the reviewers can address them easily. The color of each reviewer is next to the reviewer number.
Reviewer 1 (Teal)
1.- Figure 1 resolution is very less
Correction: Figure 1 resolution was updated accordingly
2.- Explain novelty of your work
Correction: We agree that the novelty of the work was not properly highlighted, almost all the related works include the “glucose” feature the model, nevertheless, this research is focused only on features not related to the glucose, as a way to create a non invasive biomarker, the ML process for the model generation also uses the ensemble approach to generate a robust model with a very good performance, this process was tested on a mexican database.
To highlight the novelty the following actions were made:
The following text was added on lines 167-176: “This model seeks to have exclusively features non-related with glucose. With the use of multivariate models, which is, the use of multiple features (Non-invasive and non-related glucose features in this work) applied into a model so correlations or patterns can be found, used for the detection of T2DM. The proposed ML ensemble model focuses on classifying the 1787 patients with an input of the 48 availables in the processed database. The dataset for this study is acquired from “Unidad de Investigación Médica en Bioquímica, Centro Médico Nacional Siglo XXI, IMSS”, with the information of Mexican patients. The database contains anthropometric data, medical treatment, complications, lipid profile and blood pressure. It is worth mentioning that there are no public datasets with this type of patients”,.
Comparison table against other approaches was added on table 14, proper discussion was added to section 4, on lines 405-407.
3.- What you mean non dichotomous data, kindly explain in your paper.
Correction: The non dichotomous data refers to all the data that is not the output column, correction was added in line 200-201
4.- What you infer in Z score from your work, please include.
Correction: The Z-score is the normalization process that transform the different range features into a same range features, for this we used mean 0 and SD =1, correction was added on line 227
5.- How you used tuning parameters as variable parameters.
Correction: We are assuming that you refer to the tuning parameters of the algorithms such as the “C” and “gamma” for the SVM algorithm, and similar for lasso and such, in order to avoid bias for a specific algorithm we use the default parameters in the R - implementations.
6.- Which kernel are used in SVM.
Correction: For this research we used the “RBF kernel”, we added the kernel specification on line 303
7.- Kindly compare your results with minimum three recent papers
Correction: we agree, Comparison table against other approaches was added on table 14, on line 407, proper discussion was added to section 4.

Reviewer 2 Report
The manuscript entitled “Ensemble Model for the Detection of Type 2 Diabetes in Mexican Population” presents a good idea of research for the type-2 diabetic patients in Mexico. The presented manuscript is related to one of the applications of artificial intelligence in healthcare and medicine. But the manuscript structure needs improvement. The list also includes many shortcomings related to the experiments, methodology, grammar, and the results of this research work. I will recommend the major revisions with resubmission, following major steps that need to be evaluated for the better contribution and novelty claimed by the authors.
· Authors only focused on the machine learning algorithms and did not mention the ensembling technique (Max voting) used in the abstract of the manuscript.
· In the abstract one abbreviation has been done in capital letters but others in small letters i.e. (ANN) Artificial Neural Network, and for (ML) for machine learning. It should be similar to the whole manuscript (small letters preferred).
· The Fig. 1. Illustrates the proposed methodology. The authors again did not mention which ensembling technique is used in the manuscript, nor another basic step in the data preprocessing, handling missing values, feature selection part etc. not reported in the figure. It’s a suggestion to re-draw this fig with all details for reference see the following article figure 1, get the idea from the following article and cite in your research, “Effective Voting Ensemble of Homogenous Ensembling with Multiple Attribute-Selection Approaches for Improved Identification of Thyroid Disorder”.
· This ensembling method is very widely used by many authors, authors did not mention either soft or hard voting used in this ensembling. Again for reference see the above-mentioned article, figure 4.
· In the introduction, many small paragraphs are stated by authors, which can lose the interest of the readers, these can be reduced to less number of paragraphs, for a better presentation of the manuscript.
· In implementation (2.12) authors forget to add the ensembling part.
· In features selection, LASSO is used but how many features were selected before and after the attribute selection (very important detail) is again missed in the manuscript.
· In performance evaluation metrics only AUC is used, why authors do not use other metrics, accuracy, recall, f1-score, etc.
· The results (AUC) of SVM are much better than the ensembling model, why do authors mainly focus on ensembling in the title?
· In conclusion, discussion and other important sections of the manuscript, multiple small numbers of paragraphs written. Such an approach can divert the attention of the reader from actual details of the research to other texts.
· Many crucial details of hyperparameters tuning from sections 2.8, 2.9, 2.10, and 2.11 were also missed in the manuscript.
· In line 244 what do you mean by 10-fold crossover?
· Line 241, 242..the method of citing documents by giving the full name of researchers i.e. Asif Hassan Syed and Tabraej Khan is not the proper way to cite in the manuscript. Please go through multiple published articles in the “Healthcare” journal.
· The writing of the manuscript needs significant improvement.
· Authors just compare with different machine learning techniques but did not compare their results in a table and also other related existing studies.
· The result and discussion section are not elaborately discussed.
Author Response
List of changes
The paper reviewed have text color changes so the reviewers can address them easily. The color of each reviewer is next to the reviewer number.
Reviewer 2 (Magenta)
1.- Authors only focused on the machine learning algorithms and did not mention the ensembling technique (Max voting) used in the abstract of the manuscript.
Correction: We agree, in order to avoid confusion for the reader, the hard voting ensemble was added to the abstract and materials and methods section.
2.- In the abstract one abbreviation has been done in capital letters but others in small letters i.e. (ANN) Artificial Neural Network, and for (ML) for machine learning. It should be similar to the whole manuscript (small letters preferred).
Correction: Thank you, we agree, we revised the abbreviations for the whole manuscript, for the suggest lines, were changed on line 5 and 6
3.- The Fig. 1. Illustrates the proposed methodology. The authors again did not mention which ensembling technique is used in the manuscript, nor another basic step in the data preprocessing, handling missing values, feature selection part etc. not reported in the figure. It’s a suggestion to re-draw this fig with all details for reference see the following article figure 1, get the idea from the following article and cite in your research, “Effective Voting Ensemble of Homogenous Ensembling with Multiple Attribute-Selection Approaches for Improved Identification of Thyroid Disorder”.
Correction: Thank you, figure 1 was updated accordingly, suggested reference was added to the related works in lines 183-184.
4.- This ensembling method is very widely used by many authors, authors did not mention either soft or hard voting used in this ensembling. Again for reference see the above-mentioned article, figure 4.
Correction: The hard voting was added in the abstract and materials and methods to avoid confusion, title was changed to: “Hard Voting Ensemble Approach for the Detection of Type 2 Diabetes in Mexican Population with Non-glucose related Features” to affirm the type of ensemble used.
5.- In the introduction, many small paragraphs are stated by authors, which can lose the interest of the readers, these can be reduced to less number of paragraphs, for a better presentation of the manuscript.
Correction: The manuscript was rewritten to avoid such small paragraphs.
6.- In implementation (2.12) authors forget to add the ensembling part.
Correction: Thank you, changes were made on line 326
7.- In features selection, LASSO is used but how many features were selected before and after the attribute selection (very important detail) is again missed in the manuscript.
Correction: Thank you, we agree the LASSO methodology reduced from 22 a 12 features, changes were made to lines: 242-243
8.- In performance evaluation metrics only AUC is used, why authors do not use other metrics, accuracy, recall, f1-score, etc.
Correction: We updated the metrics used to assess the performance, suggested metrics were added on lines: 334-340, equations from 8 to 15 and Tables 6, 8, 10 and 12
9.- The results (AUC) of SVM are much better than the ensembling model, why do authors mainly focus on ensembling in the title?
Correction: We agree with the higher AUC of the SVM, nevertheless, the SVM performance varies according to the data partition on several experiments the performance ranged from the 0.87 - 0.928 while the ensemble model remained with very low variance, ranging from 0.89 - 0.905, thus we selected the best scenario, in this case the best for the SVM and the best for the Ensemble (using the same seed for both in order to avoid bias to a specific algorithm), proper discussion was added line 412-415
10.- In conclusion, discussion and other important sections of the manuscript, multiple small numbers of paragraphs written. Such an approach can divert the attention of the reader from actual details of the research to other texts.
Correction: The manuscript was rewritten to avoid such small paragraphs.
11.- Many crucial details of hyperparameters tuning from sections 2.8, 2.9, 2.10, and 2.11 were also missed in the manuscript.
Correction: We specify the hyperparameters by default, these were used in the models implemented.
12.- In line 244 what do you mean by 10-fold crossover?
Correction: We refer to cross-validation k =10, changes were made on line 277
13.- Line 241, 242..the method of citing documents by giving the full name of researchers i.e. Asif Hassan Syed and Tabraej Khan is not the proper way to cite in the manuscript. Please go through multiple published articles in the “Healthcare” journal.
Correction: Thank you, citation was revised, on line 84
14.- The writing of the manuscript needs significant improvement.
Correction: All text has been reviewed according to the reviewers puntual specifications.
15.- Authors just compare with different machine learning techniques but did not compare their results in a Table and also other related existing studies.
Correction: Comparison table against other approaches was added on table 14, proper discussion was added to section 4, on lines 405-407.
16.- The result and discussion section are not elaborately discussed.
Correction: Discussion was updated in line 367-370, 405-407, 408-411, 412-415, Comparison table against other approaches was added on table 14, proper discussion was added to section 4, on line 396.

Reviewer 3 Report
In this manuscript, the authors proposed an ensemble model for the detection of type 2 diabetes in the Mexican population. The topic, in general, seems interesting; however, this work doesn't address it well-i.e., there are several issues that the authors should work on to improve the manuscript. - The innovation is insufficient. Similar work has been done before. The proposed algorithm should be compared to other alternative approaches. - Figure 1 is not intuitive, it is recommended to use a standard flow chart. - The authors claimed that “a case elimination approach is used, this approach consists of working only with the observations that have complete data for all variables and discarding all others, removing all out-of-scope or unusable features ...” Why use this method? The author should have further clarified. - The authors claimed that “ The developed LASSO implementation solves the problem...”. However, there is no explanation of why chose LASSO. The authors have not compared the LASSO with other similar feature selection methods regarding the differences, the similarities, etc. - The authors just reviewed a limited number of existing works. The authors are suggested to provide more existing works. For example, some well-known and highly cited works such as 10.1109/ACCESS.2021.3091622, 10.1093/jamia/ocaa283 and 10.1166/jmihi.2019.2582. - The evaluation metrics are not explained in detail. Also, the metrics must be written in the form of equations. Although famous, some readers may not know. - Can the author provide some discussion about how the data was split into test/train/validation sets? Why does the experiment use 75% for training and 25% for testing? - Many parts of the paper do not read well. The authors are advised to review the paper to improve its readability carefully.Author Response
List of changes
The paper reviewed have text color changes so the reviewers can address them easily. The color of each reviewer is next to the reviewer number.
Reviewer 3 (Purple)
1.- The innovation is insufficient. Similar work has been done before. The proposed algorithm should be compared to other alternative approaches.
Correction: We agree, this was also highlighted by the reviewer 1 and 2.
The novelty of the work was not properly highlighted, almost all the related works include the “glucose” feature the model, nevertheless, this research is focused only on features not related to the glucose, as a way to create a non invasive biomarker, the ML process for the model generation also uses the ensemble approach to generate a robust model with a very good performance, this process was tested on a mexican database.
The following text was added on lines 167-176: “This model seeks to have exclusively features non-related with glucose. With the use of multivariate models, which is, the use of multiple features (Non-invasive and non-related glucose features in this work) applied into a model so correlations or patterns can be found, used for the detection of T2DM. The proposed ML ensemble model focuses on classifying the 1787 patients with an input of the 48 availables in the processed database. The dataset for this study is acquired from “Unidad de Investigación Médica en Bioquímica, Centro Médico Nacional Siglo XXI, IMSS”, with the information of Mexican patients. The database contains anthropometric data, medical treatment, complications, lipid profile and blood pressure. It is worth mentioning that there are no public datasets with this type of patients”,.
Comparison table against other approaches was added on table 14, proper discussion was added to section 4, on lines 405-407.
2.- Figure 1 is not intuitive, it is recommended to use a standard flow chart.
Correction: Figure 1 was updated according to the reviewer 2 suggestions.
3.- The authors claimed that “a case elimination approach is used, this approach consists of working only with the observations that have complete data for all variables and discarding all others, removing all out-of-scope or unusable features ...” Why use this method? The author should have further clarified.
Correction: Subjects with missing data and such, were removed in order to avoid bias to an specific interpolator, this was possible as the number of subjects was very high thus removing some subjects did not impact on the performance data, changes were made to lines 203-224
4.- The authors claimed that “ The developed LASSO implementation solves the problem...”. However, there is no explanation of why chose LASSO. The authors have not compared the LASSO with other similar feature selection methods regarding the differences, the similarities, etc.
Correction: We based our research on the article title: “Machine Learning For Tuning, Selection, And Ensemble Of Multiple Risk Scores For Predicting Type 2 Diabetes”, by Liu, Y. et al. (2019), where a proper comparison of feature selection process is presented, here the LASSO achieved a very good performance, we agree that other methods may improve the performance such as genetic algorithms, or boruta, nevertheless, our approach is targets the ensemble procedure rather than the feature selection process. changes to lines 233-242 and lines 244-246 were made.
5.- The authors just reviewed a limited number of existing works. The authors are suggested to provide more existing works. For example, some well-known and highly cited works such as 10.1109/ACCESS.2021.3091622, 10.1093/jamia/ocaa283 and 10.1166/jmihi.2019.2582.
Correction: Suggested references were added in lines 63-66, also changes to lines 111-152 and Table 14 were added
6.- The evaluation metrics are not explained in detail. Also, the metrics must be written in the form of equations. Although famous, some readers may not know.
Correction: We agree, the reviewer 2 also suggested this, we add these correction on lines: 334-340, equations from 8 to 15 and Tables 6, 8, 10 and 12
7.- Can the author provide some discussion about how the data was split into test/train/validation sets? Why does the experiment use 75% for training and 25% for testing?
Correction: The 75% - 25% is the one of the main data partition scheme, as presented by Moreno-Torres, J.G et al. (2012) on “. Study on the Impact of Partition-Induced Dataset Shift on -Fold Cross-Validation”, lines 367-370 were rewrited
8.- Many parts of the paper do not read well. The authors are advised to review the paper to improve its readability carefully.
Correction: All text has been reviewed according to the reviewers puntual specifications.

Reviewer 4 Report
Comments
1 – Line 87. We have “Principle Component Analysis feature selection”. I suggest to change to “Principal Component Analysis-based dimensionality reduction”. Notice that PCA is not a feature selection method since it performs dimensionality reduction by feature reduction and not by feature selection.
2 – Figure 1 is too small. Please, increase the size of this figure. The contents of the figure should be changed in such a way the test data is applied to the learned ANN, SVM, and GLM models (after training).
3 – Please, do not use symbols with double meaning. On equation (1), \sigma is standard deviation. Later, in equation (5) it represents the sigmoid function. Later in equation (6), we have again the \sigma symbol.
4 – On the experimental results (tables 4 to 7), the authors may consider displaying the confusion matrices in the standard way (rows are the true labels and columns are the predicted labels).
5 – I suggest to add a discussion on the false positive rate and the false negative rate, for the classifiers under evaluation.
Writing
Line 9
10 K-fold
->
10-fold
Line 16
mellitus detection)
->
mellitus detection.
Line 22
The International Diabetes Foundation (IDF)
->
According to the International Diabetes Foundation (IDF),
Line 27
amputation[4].
->
amputation [4].
Line 65
covid-19
->
COVID-19
From line 72 to line 78, we have a single sentence, from “The use of ML models……79,7%[12]”. This is a long sentence. Please, break it into two or three smaller sentences.
Line 87
Principle Component Analysis
->
Principal Component Analysis
5 K-fold cross-validation
->
5-fold cross-validation
Line 114
of this features
->
of these features
From line 119 to line 125, we have a single sentence, from “Kocbek…..” until “[17].” This is a long sentence. Please, break it into two or three smaller sentences. Please avoid long sentences on the entire paper.
Line 120
80%-20%
->
with 80%-20%
and testing respectively
->
and testing, respectively
Line 142
on clinical data; Using these data
->
on clinical data. Using these data
Line 154
how it is validated, Section III
->
how it is validated. Section III
Line 155
output graphs, finally, Section IV
->
output graphs. Finally, Section IV
Line 168
R-2011-785-018 and includes
->
R-2011-785-018. It includes
Line 170
In Figure 2 it is shown the
->
Figure 2 shows the
Line 177
approach is used, this approach consists
->
approach is used. This approach consists
Line 181
consecutive number respectively.
->
consecutive number, respectively.
Before equation (1)
formula[19]:
->
formula [19]:
Line 215
the lasso penalty
->
the LASSO penalty
Line 225
, a \alpha=0.5 tends
->
, \alpha=0.5 tends
Line 231
mismatching; In order
->
mismatching. In order
Line 239
using cross validation, the goal of this split
->
using cross validation. The goal of this split
Line 244
Please revise the entire sentence, because is not making sense:
“in the ensemble, the validation A 10-fold crossover”
Last sentence of section 2.9. We have “1,and0whenxissmaller[20]”. Please, correct this.
Line 270
where x y x’
->
where x and x’
Line 285
(table 1)
->
(Table 1)
Line 292
in table 4
->
in Table 4.
Line 301
shown in table 6
->
shown in Table 6
Line 311
been the SVM model a 2.3% better
->
with the SVM model being 2.3% better
Line 312
shown in the figure table 3.
->
shown in Table 3.
Line 338
Decease Control
->
Disease Control
this cases
->
these cases
the risk of develop
->
the risk to develop
Line 385
with same or superior
->
with the same or superior
Author Response
List of changes
The paper reviewed have text color changes so the reviewers can address them easily. The color of each reviewer is next to the reviewer number.
Reviewer 4 (Blue)
1.- Line 87. We have “Principle Component Analysis feature selection”. I suggest to change to “Principal Component Analysis-based dimensionality reduction”. Notice that PCA is not a feature selection method since it performs dimensionality reduction by feature reduction and not by feature selection.
Correction: We agree with this, changes were reflected on line 79-80
2.- Figure 1 is too small. Please, increase the size of this figure. The contents of the figure should be changed in such a way the test data is applied to the learned ANN, SVM, and GLM models (after training).
Correction: Figure 1 was updated
3.- Please, do not use symbols with double meaning. On equation (1), \sigma is standard deviation. Later, in equation (5) it represents the sigmoid function. Later in equation (6), we have again the \sigma symbol.
Correction: Thank you very much, indeed the symbols need to be standardized, equation 5 and line 296 was updated
4.- On the experimental results (Tables 4 to 7), the authors may consider displaying the confusion matrices in the standard way (rows are the true labels and columns are the predicted labels).
Correction: Confusion matrix was updated, furthermore Table 5 is given as structure and the tables 7,9,11 and 13 have been modified accordly.
5.- I suggest to add a discussion on the false positive rate and the false negative rate, for the classifiers under evaluation.
Correction: suggested performance metrics were added, reviewer 2 also suggested the addition of other performance metrics, lines 408-411 were rewrited. Equations from 8 to 15 and Tables 6, 8, 10 and 12 were updated
Writing
Line 9
10 K-fold
->
Correction: Line 10
10-fold
Line 16
mellitus detection)
->
Correction: Line 16
mellitus detection.
Line 22
The International Diabetes Foundation (IDF)
->
Correction: Line 22-23
According to the International Diabetes Federation (IDF),
Line 27
amputation[4].
->
Correction: Line 26
amputation [4].
Line 65
covid-19
->
Correction: Line 56
COVID-19
From line 72 to line 78, we have a single sentence, from “The use of ML models……79,7%[12]”. This is a long sentence. Please, break it into two or three smaller sentences.
Correction: Line 69
Line 87
Principle Component Analysis
->
Correction: Line 79
Principal Component Analysis
5 K-fold cross-validation
->
Correction: Line 80
5-fold cross-validation
Line 114
of this features
->
Correction: Line 102
of these features
From line 119 to line 125, we have a single sentence, from “Kocbek…..” until “[17].” This is a long sentence. Please, break it into two or three smaller sentences. Please avoid long sentences on the entire paper.
Correction: Line 101-102
Line 120
80%-20%
->
Correction: Line 101
with 80%-20%
and testing respectively
->
Correction: Line 106
and testing, respectively
Line 142
on clinical data; Using these data
->
Correction: Line 166
on clinical data. Using these data
Line 154
how it is validated, Section III
->
Correction: Line 179
how it is validated. Section III
Line 155
output graphs, finally, Section IV
->
Correction: Line 180
output graphs. Finally, Section IV
Line 168
R-2011-785-018 and includes
->
Correction: Line 193
R-2011-785-018. It includes
Line 170
In Figure 2 it is shown the
->
Correction: Line 195
Figure 2 shows the
Line 177
approach is used, this approach consists
->
Correction: This line was modified
Line 181
consecutive number respectively.
->
Correction: Line 206
consecutive number, respectively.
Before equation (1)
formula[19]:
->
Correction: Before equation (1)
formula [19]:
Line 215
the lasso penalty
->
Correction: Line 250
the LASSO penalty
Line 225
, a \alpha=0.5 tends
->
Correction: Line 259
, \alpha=0.5 tends
Line 231
mismatching; In order
->
Correction: Line 264-265
mismatching. In order
Line 239
using cross validation, the goal of this split
->
Correction: Line 272
using cross validation. The goal of this split
Line 244
Please revise the entire sentence, because is not making sense:
“in the ensemble, the validation A 10-fold crossover”
Correction: Line 277
Last sentence of section 2.9. We have “1,and0whenxissmaller[20]”. Please, correct this.
Correction: Line 298
Line 270
where x y x’
->
Correction: Line 308
where x and x’
Line 285
(Table 1)
->
Correction: Line 331
(Table 1)
Line 292
in Table 4
->
Correction: Line 347
in Table 7.
Line 301
shown in Table 6
->
Correction: Line 354
shown in Table 11
Line 311
been the SVM model a 2.3% better
->
Correction: Line 362
with the SVM model being 2.3% better
Line 312
shown in the figure Table 3.
->
Correction: Line 363
shown in Table 3.
Line 338
Decease Control
->
Correction: Line 386
Disease Control
this cases
->
Correction: Line 387
these cases
the risk of develop
->
Correction: Line 387
the risk to develop
Line 385
with same or superior
->
Correction: Line 435
with the same or superior

Round 2
Reviewer 1 Report
No Comments.
Reviewer 2 Report
The authors have successfully incorporated all my points and concerns, therefore this manuscript can be accepted in its current form.
Reviewer 3 Report
I think the authors have put so many efforts on this revised manuscript. I was reading comments of other reviewers and comparing with the original submitted manuscript. The manuscript was deeply revised. It was interesting to read responses to other reviewers that showed deep revision procedure and addressing concerns about the manuscript. In my point of view, the manuscript is ready to be accepted.